# Bioavailability and Metabolic Fate of (Poly)phenols from Hull-Less Purple Whole-Grain Barley in Humans

**DOI:** 10.3390/nu17193086

**Published:** 2025-09-28

**Authors:** María-Engracia Cortijo-Alfonso, Silvia Yuste, Mariona Martínez-Subirà, Marian Moralejo, Carme Piñol-Felis, Alba Macià, Laura Rubió-Piqué

**Affiliations:** 1Department of Food Technology, Engineering and Science, University of Lleida-Agrotecnio CERCA Center, Av. Alcalde Rovira Roure 191, 25198 Lleida, Spain; engracia.cortijo@udl.cat (M.-E.C.-A.); silvia.yuste@udl.cat (S.Y.); mariona.martinez@udl.cat (M.M.-S.); marian.moralejo@udl.cat (M.M.); laura.rubio@udl.cat (L.R.-P.); 2Department of Medicine and Surgery, University of Lleida, 25198 Lleida, Spain; carme.pinyol@udl.cat; 3Institut de Recerca Biomèdica de Lleida Fundació Dr. Pifarré, IRBLleida, 25198 Lleida, Spain

**Keywords:** barley (poly)phenols, bioavailability, metabolism, postprandial study, UPLC-MS/MS

## Abstract

**Background and Objectives:** Anthocyanin-rich barley varieties have recently gained attention due to their high (poly)phenolic content and potential health benefits, yet human data on their bioavailability remain scarce. This study aimed to characterize the absorption, metabolism, and excretion of (poly)phenolic compounds from a novel hull-less purple whole-grain barley (WGB) genotype. **Methods:** Eleven healthy volunteers consumed 140 g of purple WGB biscuits, and plasma and urine samples were collected over 6 h and 48 h, respectively. **Results:** UPLC-MS/MS analysis revealed a broad range of metabolites, with 11 (poly)phenolic compounds identified in plasma and 80 in urine. The biscuits were particularly rich in flavones (217 mg/140 g, mainly chrysoeriol derivatives), followed by hydroxycinnamic acids (~54 mg, mainly 4′-hydroxy-3′-methoxycinnamic acid), anthocyanins (44.8 mg), and flavan-3-ols (16.8 mg). In plasma, glycosylated anthocyanins and flavone conjugates (e.g., peonidin-3-O-glucuronide, chrysoeriol-*O*-glucuronide) were detectable within 1–2 h, consistent with early absorption. In contrast, microbial-derived catabolites—including valerolactones, phenylacetic and benzoic acids—were mainly excreted in urine between 8 and 24 h, reaching concentrations above 1000 nM. **Conclusions:** These findings provide novel insights into the bioavailability and metabolic fate of barley (poly)phenols, supporting their potential contribution to host and gut health. As a proof-of-concept study, it complements the limited data available from pigmented cereals and underscores the need for validation in larger cohorts.

## 1. Introduction

The health benefits associated with whole grain consumption are well-documented and have been consistently linked to a reduced risk of chronic diseases, including cardiovascular disease, type 2 diabetes, and certain cancers [1,2]. These protective effects are attributed not only to the high content of dietary fiber but also to the presence of a wide range of bioactive compounds, particularly (poly)phenolic compounds.

Barley is a nutrient-rich whole grain, particularly high in soluble fiber (mainly β-glucans) and (poly)phenolic compounds. Although its consumption is relatively limited in many regions, barley has gained attention due to its antioxidant, anti-inflammatory, and cholesterol-lowering properties. However, in whole grain products, (poly)phenolic compounds are frequently bound—either covalently or non-covalently—to the dietary fiber matrix, which substantially limits their absorption in the upper gastrointestinal tract. As a result, a large proportion reach the colon intact, where they are subject to microbial enzymatic activity (e.g., β-glucosidases, esterases). This microbial metabolism releases simpler (poly)phenolic metabolites, which may extend their systemic availability and enhance their bioactivity, underscoring the key role of colonic transformation in mediating the health benefits of whole grain barley intake [3].

Pigmented cereals, such as black [4] and purple barley [5,6], as well as purple [5,7] and blue wheat [5], have attracted increasing research interest due to their high contents of anthocyanins and other (poly)phenolic compounds that exhibit strong antioxidant and anti-inflammatory properties. Despite the increasing interest in these pigmented lines, there is still limited evidence regarding the bioavailability and metabolic fate of their (poly)phenols in humans.

To date, only two human studies have assessed the absorption and excretion of (poly)phenols from pigmented cereals, specifically wheat [5,7] and barley [5]. These studies focused on the urinary excretion of (poly)phenolic metabolites after an acute intake of either a cooked whole-grain porridge [5] or bran-enriched purple wheat bars and crackers [7]. However, both studies present important limitations, including a small sample size (*n* = 3), restricted sampling at a single 4 h urine collection point, and low metabolite coverage [5], or reliance on HPLC-DAD as the analytical technique [7]. Such limitations underscore the need for more comprehensive investigations by employing extended sampling periods and advanced analytical techniques.

In response, the present study provides a comprehensive assessment of the (poly)phenolic profile of biscuits made with purple whole grain barley and investigates their metabolic fate in humans. Using a targeted UPLC-MS/MS approach, we identified and quantified (poly)phenolic metabolites in plasma and urine collected at multiple time points. To our knowledge, this is the first human study to characterize the absorption, metabolism, and excretion of (poly)phenols from processed pigmented barley, offering novel insights into their bioavailability and health-relevant transformations.

In this study, the metabolite nomenclature follows the recommendations of Kay et al., 2020 [8], and Curti et al., 2025 [9], with all referenced literature adjusted accordingly.

## 2. Materials and Methods

### 2.1. Chemicals and Reagents

The commercial standards 1,2-dihydroxybenzene (known as catechol), 4-hydroxybenzoic acid, 3,4-dihydroxybenzoic acid (known as protocatechuic acid), 4-hydroxy-3-methoxybenzoic acid (known as vanillic acid), 4-hydroxy-3,5-dimethoxybenzoic acid (known as syringic acid), 3-(4′-hydroxyphenyl)propanoic acid, 3-(4′-hydroxyphenyl)propanoic acid, 3-(3′,4′-dihydroxyphenyl)propanoic acid and 3-(4-hydroxy-3-methoxyphenyl)propanoic acid (known as dihydroferulic acid) were from Sigma-Aldrich (St. Louis, MO, USA). Cyanidin-3-*O*-glucoside, 4′-hydroxycinnamic acid, 3′,4′-dihydroxycinnamic acid (known as caffeic acid), 4′-hydroxy-3′-methoxycinnamic acid (known as ferulic acid), 4-hydroxy-3,5-dimethoxycinnamic acid (known as sinapic acid), catechin, chrysoeriol (luteolin-3′-methyl ether), and luteolin-7-*O*-glucuronide were from Extrasynthese (Genay, France).

Acetonitrile (HPLC-grade) was from Romil (Tecknokroma, Barcelona, Spain). Methanol (HPLC-grade), phosphoric acid (85%), glacial acetic acid (99.8%), formic acid, sodium hydroxide (NaOH) and hydrochloric acid (HCl) (37%) were from Scharlau S.L. (Barcelona Spain). Ultrapure water was obtained from a Mili-Q water purification system (Millipore Corp., Bedford, MA, USA).

Stock solutions of (poly)phenolic compounds were prepared by dissolving each standard compound in methanol at a concentration of 1000 mg/L and stored in a dark flask at −30 °C.

### 2.2. Preparation of Whole-Grain Barley Biscuits

Home-made biscuits were produced from purple-whole grain barley (WGB) following the procedure described previously [10], which is based on AACC method 10–50.05 with minor modifications. The formulation included 51% WGB flour (dry basis), along with sugar, vegetable margarine, dextrose solution, tartaric acid, and leavening agents. After mixing and shaping, the biscuits were baked at 200 °C for 10 min. Finished products were vacuum-packed and stored at 4 °C until used in the chemical analyses and human intake study.

The extraction and quantitative determination of free and bound (poly)phenolic compounds from the purple WGB biscuits were performed according to the methodology described in our previous study [10]. The nutritional composition of the biscuits consumed in the intervention is detailed in Appendix A.

### 2.3. Subjects and Study Design

The present analysis is part of an acute, randomized, crossover trial comparing the absorption and metabolism of (poly)phenols from two barley-based biscuits: one made with whole grain barley (WGB) and another made exclusively from barley bran. In this article, we report only the results obtained after the consumption of the WGB biscuit.

A total of 11 healthy participants (6 women and 5 men, aged 25–45 years, BMI 20–25 kg/m^2^) were recruited through advertisements posted at the local hospital and affiliated research centers. All volunteers were non-smokers and not taking any chronic medication or antioxidant supplements. Exclusion criteria included: age below 18 years, age above 60 years, pregnancy or lactation, chronic illness, recent antibiotic use (within the past 4 months), regular medication or dietary supplement intake, smoking, and excessive alcohol consumption (>80 g/day).

Prior to enrolment, subjects were placed on a restricted low-(poly)phenol diet and to abstain from consuming (poly)phenol-rich foods (such as coffee, fruits, vegetables, chocolate, tea, and red wine) for 3 days. On the study day, following an overnight fast, each subject consumed 140 g of WGB biscuits. Appendix A shows a graphical overview of the study design.

Venous blood samples were collected at baseline (t = 0 h) and at 1, 2, 4, and 6 h post-consumption using 6 mL EDTA-containing Vacutainer tubes (Becton Dickinson, Franklin Lakes, NJ, USA). Plasma was obtained by centrifugation at 8784× *g* for 15 min (Hettich, Tuttlingen, Germany), aliquoted, and stored at −80 °C until analysis.

Baseline urine samples were obtained from the 24 h period prior to the intervention. On the study day, urine was collected at 0–2 h, 2–4 h, 4–8 h, and 8–24 h intervals. A final urine sample was also collected between 24 and 48 h to assess delayed metabolite excretion.

The protocol received approval from the Ethical Committee of Human Clinical Research at the Arnau Vilanova University Hospital, Lleida, Spain (CEIC-2387, 14 December 2020). Written informed consent was obtained from each participant prior to study participation.

### 2.4. Biological Samples Pretreatment

#### 2.4.1. Plasma Analysis

For the analysis of the main circulating (poly)phenolic metabolites in plasma samples, microElution solid-phase extraction (μSPE) was performed as described in our previous studies [10]. Briefly, μElution plates (OASIS HLB 2 mg, Waters, Milford, MA, USA) were first conditioned with 250 μL of methanol and then equilibrated with 250 μL of 0.2% acetic acid to activate the sorbent to sample loading. A volume of 350 μL of venous plasma was mixed with 350 μL of 4% phosphoric acid and subsequently centrifuged at 8784× *g* for 10 min and 4 °C. Then, the supernatant was transferred to the μElution cartridge. Subsequently, the loaded micro-cartridges were cleaned-up with 200 μL of Milli-Q water and 200 μL of 0.2% acetic acid. Elution of the retained (poly)phenolic metabolites was achieved using two sequential 50 μL aliquots of methanol, after which 2.5 μL of the resulting eluate was directly introduced into the UPLC-MS/MS system.

#### 2.4.2. Urine Analysis

For the analysis of the main urinary excreted (poly)phenolic metabolites, microElution solid-phase extraction (μSPE) was also carried out as previously described in our studies [11]. Briefly, μElution plates (OASIS HLB 2 mg, Waters, Milford, MA, USA) were conditioned with 250 μL of methanol and subsequently equilibrated with 250 μL of 0.2% acetic acid. Urinary samples (100 μL) were mixed with 100 μL of 4% phosphoric acid and centrifuged at 8784× *g* for 10 min at 4 °C, after with the supernatant was loaded onto the micro-cartridge. The retained (poly)phenolic metabolites were then eluted with two sequential 50 μL aliquots of methanol, and 2.5 μL of the eluate was directly injected into the UPLC-MS/MS system.

### 2.5. Analysis of (Poly)phenolic Metabolites by UPLC-MS/MS

(Poly)phenolic compounds and their metabolites were quantified in plasma and urine samples using an AcQuity UPLC system coupled to a triple quadrupole detector (TQD) mass spectrometer (Waters, Milford, MA, USA). Separation was performed on a BEH C_18_ column (100 mm × 2.1 mm id, 1.7 μm) maintained at 30 °C, with a flow rate of 0.3 mL·min^−1^. The mobile phase and the elution gradient were the same as those reported in our previous studies [10,11].

Tandem MS analyses were carried out using a triple quadrupole detector (TQD) mass spectrometer (Waters, Milford, MA, USA) equipped with a Z-spray electrospray interface. Ionization was performed in positive mode [M − H]^+^ for the analysis of anthocyanins and the flavones (methyl) luteolin-*O*-glucuronide, and in negative mode [M − H]^−^ for the rest of (poly)phenolic compounds. Data acquisition was conducted using selected reaction monitoring (SRM), and the ion source parameters were set as previously reported [10,12].

Two SRM transitions were selected for each compound: the most sensitive transition was used for quantification, and the second for confirmation. Appendix A shows the SRM transitions for quantification, along with the corresponding cone voltages and collision energies for each (poly)phenolic metabolite. The dwell time for each transition was set at 30 ms, and data acquisition was performed using MassLynx 4.1 software. Due to the unavailability of commercial standards for certain (poly)phenolics and its metabolites, some of these compounds were tentatively quantified using the calibration curve of the parent compound or a structurally similar (poly)phenolic standard. Appendix A also shows the standard used for the quantification of each compound.

### 2.6. Statistical Analysis

The results are presented as mean values ± standard deviation (SD) for the determination of the (poly)phenolic compounds in the WGB biscuits. Regarding the analysis of biological samples, the results are expressed as mean values ± standard error of the mean (SEM) for the generated (poly)phenolic metabolites in plasma and urine samples.

Quantitative data were analyzed using a paired Student *t*-test (ANOVA one-way) to assess significant differences in the mean levels of the main circulating barley (poly)phenolic metabolites. The analyses included both plasma samples at various post-intake times (0, 1, 2, 4 and 6 h) following the acute intake of WGB biscuits, and urine samples collected at different time interval points (0–48 h) post-consumption of the WGB biscuits.

Significant differences were considered at the level of *p* < 0.05. All the statistical analyses were carried out using STATGRAPHICS Plus 5.1 (Manugistics Inc., Rockville, MD, USA).

## 3. Results and Discussion

### 3.1. Characterization of (Poly)phenolic Compounds in Barley Biscuits

Table 1 presents the (poly)phenolic composition of the WGB biscuits, which were analyzed to quantify both free and bound forms. (Poly)phenols in cereals like barley exist mainly in two forms: (a) soluble/free forms, either as aglycones or conjugates (e.g., glucosides, esters), and (b) bound forms, covalently linked to the cell wall matrix—predominantly arabinoxylans.

The total (poly)phenolic content in 140 g of WGB biscuits was 359 ± 32.2 mg. Of this, 78% was found in the free fraction, while 21% was associated with the fiber-bound fraction. This distribution reflects a notable shift towards free form compared to the original flour, where bound phenolics typically predominate [11]. This change is likely due to thermal processing during baking, which promotes the partial release of (poly)phenols from the dietary fiber matrix—mainly arabinoxylans [3].

Among the free (poly)phenolic compounds, flavones were the most abundant class, comprising 77% of the total free phenolics (217 ± 33.0 mg per 140 g). Chrysoeriol-*O*-glucuronide was the predominant flavone, representing over half of the total flavone content. This compound, along with luteolin and its derivatives, has recently been identified as characteristic of pigmented hull-less barley lines, particularly those with black or purple pigmentation [4,6,12].

Anthocyanins represented approximately 16% of the free fraction. Cyanidin-3-*O*-glucoside, along with its mono- and di-malonylated derivates, were the major anthocyanins detected. Two isomers of cyanidin mono-malonylglucoside were tentatively identified by LC-MS/MS, based on their fragmentation patterns and elution order, and these were cyanidin 3-*O*-(3″-*O*-malonyl)glucoside, and cyanidin 3-*O*-(6″-*O*-malonyl)glucoside [13]. The anthocyanins most abundant were cyanidin-3-*O*-(3″,6″-*O*-dimalonyl)glucoside (16.9 ± 0.64 mg/140 g) and cyanidin 3-O-(6″-*O*-malonyl)glucoside (12.5 ± 0.50 mg/140 g).

In the bound phenolic fraction, 4-hydroxy-3-methoxycinnamic acid (ferulic acid) and its derivatives were predominant, reaching concentrations of 70–80 mg per 140 g, and accounting for 93% of total bound (poly)phenols. This is consistent with previous reports describing this phenolic acid as the major bound phenolic acid in barley [14,15].

These results are in line with the scarce reported literature for the characterization of (poly)phenolic composition of pigmented barley cereal [6,12].

### 3.2. Plasma Appearance of Phenolic Metabolites

Plasma samples collected over the 6 h postprandial period were analyzed to evaluate the early absorption and systemic availability of (poly)phenols from purple WGB biscuits. This timeframe captures metabolites absorbed in the small intestine, including native compounds and their phase II conjugates (glucuronides, sulfates, methylated forms). Additionally, early microbial or enzymatic catabolites may also appear, reflecting initial breakdown processes occurring before colonic fermentation. This analysis allows for a comprehensive characterization of the circulating phenolic profile and helps identify compounds with potential systemic bioactivity shortly after consumption.

#### 3.2.1. Plasma Anthocyanins

Following consumption of WGB biscuits, five anthocyanin-derived compounds were identified in plasma, all corresponding to either cyanidin or peonidin derivatives (Figure 1). Although cyanidin derivatives were more abundant in barley biscuits (Table 1), peonidin derivatives—particularly peonidin-3-*O*-glucuronide—showed higher mean plasma concentrations (Figure 1F). This suggests metabolic methylation of cyanidins via catechol-*O*-methyltransferase (COMT), followed by glucuronidation by UDP-glucuronosyltransferases (UGTs), as illustrated in Appendix A, and previously described in human anthocyanin metabolism [16,17].

All compounds exhibited a similar pharmacokinetic profile, with peak plasma concentrations occurring between 1 and 2 h post-intake, indicative of rapid absorption in the upper gastrointestinal tract. No secondary peaks or delayed absorption phases were observed, suggesting limited enterohepatic recirculation (Figure 1).

Among the detected metabolites, peonidin-3-*O*-glucuronide (Figure 1F) and cyanidin-3-(6″-*O*-malonyl)glucoside (Figure 1C) stood out for their high detection frequency and systemic exposure. These compounds were present in 82% and 91% of participants, respectively, and exhibited the highest mean AUCs, with 4.48 ± 1.56 nM·h and 3.21 ± 0.55 nM·h, respectively.

In comparison with previous studies on the acute consumption of pigmented cereals, no anthocyanins were detected in plasma samples following the intake of a cooked purple whole-grain barley porridge [5], or bran-enriched purple wheat bars and crackers [7]. This absence may be attributed to the low anthocyanin content of bran-enriched purple wheat products, which was reported to be seven-fold lower [7] than that of the products (biscuits) analyzed in the present study (6.7 mg vs. 44.8 mg).

Inter-individual variability was evident both in the number of anthocyanins detected and in their plasma concentrations. Detection rates ranged from 45% to 91% depending on the compound, suggesting differences among participants in terms of absorption and metabolism.

#### 3.2.2. Rest of the (Poly)phenolic Compounds

In addition to anthocyanins, ten other (poly)phenolic metabolites were determined in plasma following the consumption of purple WGB biscuits. Among these, six metabolites exhibited early absorption with peak plasma concentrations observed at 1 or 2 h post-intake (Figure 2A–E). The most abundant was chrysoeriol-*O*-glucuronide (AUC: 853 ± 130 nM·h), followed by 4-hydroxy-3-methoxybenzoic acid-*O*-sulphate (101 ± 19.6), 3,4-dihydroxybenzoic acid-*O*-sulphate (47.7 ± 6.66), 4′-hydroxy-3′-methoxycinnamic acid-*O*-sulphate (46.7 ± 15.1), 3,4-dihydroxyphenylacetic acid-*O*-sulphate (25.4 ± 0.93), and 4′-hydroxy-3′-methoxycinnamic acid-*O*-glucuronide (9.37 ± 2.00).

Several of these compounds likely originate from conjugation of native phenolic acids present in barley, such as 4′-hydroxy-3′-methoxycinnamic acid, 4-hydroxy-3-methoxybenzoic acid, and 3,5-dimethoxy-4-hydroxybenzoic acid, via hepatic sulfation and glucuronidation.

Microbial-derived catabolites like 3,4-dihydroxybenzoic acid-*O*-sulphate and 3,4-dihydroxyphenylacetic acid-*O*-sulphate are consistent with early breakdown of cyanidin derivatives, while 4-hydroxy-3-methoxybenzoic acid-*O*-sulphate likely arises from peonidin degradation, all followed by sulfation in the liver [13]. Their early detection (1 h) suggests that deglycosylation and partial breakdown of anthocyanins may occur in the upper gut through enzymatic or non-microbial mechanisms [6].

Notably, chrysoeriol-*O*-glucuronide displayed a biphasic absorption pattern (2 h and 6 h) (Figure 2E), suggesting an initial intestinal absorption and a second release potentially due to colonic microbial action or enterohepatic recirculation, which is consistent with previous observations in capillary blood samples [10].

In contrast, four additional metabolites—dihydroxyphenyl-γ-valerolactone-*O*-sulphate (Figure 2F), dihydroxyphenyl-γ-valerolactone-*O*-glucuronide, 3-(4′-hydroxy-3′-methoxyphenyl)propanoic acid, and its sulphated form—peaked at 6 h (Appendix A). These are characteristic end-products of colonic microbial metabolism of flavan-3-ols and anthocyanins [13].

Altogether, these findings support a dual origin of circulating metabolites: early absorption in the small intestine and delayed appearance of colonic-derived catabolites, contributing to the extended and diverse plasma profile observed following WGB biscuit intake.

### 3.3. Urinary Excretion of Phenolic Metabolites

To assess the systemic fate and excretion kinetics of (poly)phenols derived from WGB biscuits, urine samples were collected over a 48 h period. A total of 80 phenolic metabolites were identified in urine, corresponding to different (poly)phenol classes, and Appendix A summarizes the cumulative urinary excretion (µmol) of these metabolite families. The following subsections detail the excretion profiles of the main compound groups, with a focus on their kinetics and origin (host or microbiota).

#### 3.3.1. Urinary Anthocyanins

Four anthocyanin metabolites were identified in urine samples following acute consumption of purple WGB biscuits. These included cyanidin-3-*O*-(3″-*O*-malonyl)glucoside, cyanidin-3-*O*-(6″-*O*-malonyl)glucoside, peonidin-3-*O*-glucoside, and peonidin-3-*O*-glucuronide. Their urinary excretion profiles are shown in Figure 3.

Cyanidin malonylglucosides showed delayed urinary excretion, with maximum concentrations observed between 4 and 24 h post-consumption, suggesting later-phase elimination or prolonged systemic circulation. In contrast, peonidin-3-*O*-glucoside and peonidin-3-*O*-glucuronide reached their excretion peaks earlier, between 2 and 4 h and 2–8 h, respectively.

Urinary excretion kinetics (Appendix A) confirmed the distinct temporal profiles of anthocyanin metabolites, with early peaks for peonidin derivatives and delayed curves for cyanidin malonylglucosides, suggesting differences in clearance or colonic metabolism. However, cumulative 48 h excretion (Appendix A) revealed similar total recovery for all four metabolites, indicating comparable systemic exposure despite inter-individual variability in excretion timing.

Interestingly, although cyanidin-3-*O*-(6″-*O*-malonyl)glucoside was the most abundant anthocyanin in both biscuits and plasma samples, its urinary levels were statistically similar to those of cyanidin-3-*O*-(3″-*O*-malonyl)glucoside. This discrepancy may stem from structural differences between the isomers. The malonyl group at the 6″ position could be more accessible to intestinal carboxyesterases or microbial esterases, enhancing its hydrolysis and subsequent metabolism. In contrast, the 3″ isomer may be more sterically hindered, resulting in lower enzymatic processing and potentially greater renal elimination. This pattern—lower urinary recovery of the 6″ isomer despite higher plasma exposure—was observed in 8 out of 11 participants

Notably, two plasma-detected compounds—peonidin-3-*O*-(3″,6″-*O*-dimalonyl)glucoside and peonidin-3-*O*-(6″-*O*-malonyl)glucoside—were not found in urine, suggesting limited renal clearance or rapid transformation into other metabolites. In contrast, peonidin-3-*O*-glucoside was detected in urine but not in plasma, potentially reflecting its direct renal excretion without significant systemic accumulation.

Compared to previous studies, our findings provide new evidence of urinary anthocyanin excretion following purple WGB intake. In contrast, the two studies available in the literature on the acute intake of pigmented cereals products reported no detectable anthocyanins in urine samples at 4 h post-consumption, either after purple whole-grain barley and wheat porridge [5], or bran-enriched purple wheat products [7]. This discrepancy is likely due to limited anthocyanin profiling and lower analytical sensitivity.

The presence of both intact glycosides and conjugates in urine suggests partial absorption and rapid clearance of anthocyanins, while delayed excretion of others points to slower colonic metabolism influenced by compound structure and host factors.

#### 3.3.2. The Rest of (Poly)phenolic Compounds

In addition to anthocyanins, a wide range of (poly)phenolic metabolites from other families were detected in urine samples. These metabolites originated from dietary flavan-3-ols, flavones, hydroxycinnamic and hydroxybenzoic acids, as well as from microbial degradation of parent compounds. Their identification provides insights into the extensive metabolic processing of WGB-derived phenolics through both host and microbial pathways. Figure 4, Figure 5, Figure 6 and Figure 7 provides an integrated overview of the proposed metabolic routes and urinary excretion kinetics of these compounds.

##### Flavan-3-ols

A total of eleven flavan-3-ols metabolites were identified in urine after acute WGB consumption (Figure 4). Among these were three phase II metabolites—(methyl) catechin-*O*-sulphate and methyl epicatechin-*O*-glucuronide—with peak excretion between 2 and 4 h post-consumption. This early appearance suggests that monomers such as catechin and epicatechin were absorbed in the small intestine and rapidly metabolized by sulfotransferase (SULT), UGT and COMT enzymes in the intestinal wall and liver.

In contrast, the remaining eight metabolites were hydroxy-γ-valerolactone derivatives, which are specific colonic catabolites formed by gut microbial metabolism of unabsorbed flavan-3-ols. These included various isomers of (di)hydroxyphenyl-γ-valerolactone in sulphate, glucuronide, and mixed sulphate–glucuronide conjugated forms (Figure 4). Urinary excretion of these compounds peaked between 8 and 24 h post-consumption, reflecting their microbial origin. 

The presence of both early-phase conjugated monomers and late-phase microbial catabolites in urine highlights the biphasic metabolism of WGB-derived flavan-3-ols. Monomers such as catechin and epicatechin, found in the biscuits (Table 1), were absorbed and cleared rapidly, whereas larger oligomers like procyanidin dimers B3 and B4 likely resisted digestion in the upper gastrointestinal tract and reached the colon, where they were transformed by the microbiota into valerolactones and further degraded into smaller phenolics (Figure 5).

##### Flavones

Eight flavone-derived metabolites were identified in urine following purple WGB consumption. These included luteolin, its methylated derivative chrysoeriol, and eriodictyol, each found in both sulphate and glucuronide conjugated variants. The urinary excretion profiles of these compounds revealed two distinct kinetic patterns, suggesting differential metabolic and absorption pathways (Figure 6).

Early excretion (4–8 h post-consumption) was observed for luteolin-*O*-sulphate and (methyl)luteolin, indicating rapid absorption in the small intestine followed by phase II metabolism—mainly sulphation and glucuronidation mediated by SULT and UGT enzymes in the enterocytes and liver.

In contrast, later excretion (8–24 h) of (methyl)luteolin-*O*-glucuronide, (methyl)luteolin-*O*-sulphate, and both conjugates of eriodictyol points to the contribution of colonic metabolism. These compounds likely result from microbial biotransformation of parent flavones in the colon, followed by absorption and conjugation prior to urinary elimination.

The metabolic conversion of luteolin to eriodictyol could also occur via phase I reduction in the small intestine. Eriodictyol may then be absorbed and conjugated or further degraded by colonic microbiota into phenylpropanoic acids—such as 3-(3,4-dihydroxyphenyl)propanoic acid [18]—and minor A-ring cleavage products like 1,3,5-trihydroxybenzene [19].

Chrysoeriol, naturally present in the WGB biscuits, may be released in the colon through microbial β-glucuronidase activity and subsequently cleaved at the C-ring, giving rise to methoxy-dihydroxybenzene. This metabolite was detected in urine as its sulphated conjugate (methoxy-dihydroxybenzene-*O*-sulphate), characterized by an [M − H]^−^ ion at *m*/*z* 219 and fragment ions at *m*/*z* 139 and 125.

As shown in Figure 6, urinary chrysoeriol levels decreased notably at 8 h post-intake, coinciding with a marked rise in chrysoeriol-*O*-glucuronide. The similar excretion profiles of chrysoeriol-*O*-glucuronide and methoxy-dihydroxybenzene-*O*-sulphate support a metabolic connection between these compounds, possibly reflecting sequential microbial and host transformations

##### Phenolic Acids

A total of thirty-three (poly)phenolic metabolites were identified in urine samples, comprising 13 hydroxybenzoic acid derivatives, and 20 hydroxycinnamic acid derivatives (Supplemental Appendix A). These compounds originated mainly from colonic microbial degradation and host-mediated phase I/II metabolism.

Among the hydroxybenzoic acids, the most abundant compounds included 4-hydroxybenzoic acid, 3,4-dihydroxybenzoic acid, their *O*-sulphated derivatives, and several methoxylated compounds. Notably, 3,4-dihydroxybenzoic acid was also detected 4 h after the acute intake of purple wheat and barley [5].

These were excreted maximally at 24 h, confirming their colonic origin. In particular, dimethoxybenzoic acid derivatives likely arose from microbial dehydrogenation and β-oxidation of 4′-hydroxy-3,5-dimethoxycinnamic acid, followed by sulphation.

Hydroxycinnamic acid metabolites—including 4′-hydroxy-3′-methoxycinnamic acid (ferulic acid), its isomer, and various conjugates (sulphate, glucuronide, glycine)—were also predominantly excreted between 8 and 24 h, consistent with release from the cereal fiber matrix in the colon and subsequent microbial transformation (Figure 7).

The phenolic 4′-hydroxy-3′-methoxycinnamic acid (ferulic acid), the most abundant hydroxycinnamic acid in cereals and mainly esterified to arabinoxylans, showed a biphasic excretion profile with peaks at 4 h and 24 h, suggesting both small intestinal absorption and colonic microbial release. The presence of 4′-hydroxy-3′-methoxycinnamic acid-*O*-sulphate in urine at 4 h and in plasma at 1 h (Figure 2) supports this dual origin. Similarly, its microbial hydrogenation product, 3-(4-hydroxy-3-methoxyphenyl)propanoic acid, also exhibited biphasic excretion (2 h and 24 h), highlighting both early host-mediated metabolism and later colonic fermentation [3,20]. In contrast, the glycine conjugate of 4′-hydroxy-3′-methoxycinnamic acid showed a single peak at 24 h, confirming its exclusive colonic microbial origin [18].

##### Non-Specific Colonic Metabolites of (Poly)phenols

Thirteen phenylacetic and phenylpropanoic acids, six 1,2-dihydroxybenzene, and five trihydroxybenzene derivatives were detected in urine following purple WGB biscuit consumption. These compounds are non-specific catabolites generated by microbial degradation of various (poly)phenolic families—anthocyanins, flavan-3-ols, flavones, and hydroxycinnamic acids—[13] as outlined in Figure 5.

Among them, 3-(3,4-dihydroxyphenyl)propanoic acid was a key microbial intermediate derived from several parent compounds. It was further dehydroxylated to 3-(4-hydroxyphenyl)propanoic acid, which underwent sulphation and was excreted in large amounts (445 ± 44.4 µmol at 48 h). Additional microbial transformations led to 3,4-dihydroxyphenylacetic acid and subsequently to 3,4-dihydroxybenzoic acid and benzene-1,2-diol derivatives via β-oxidation.

Hippuric acid was also one of the most abundant phenolic metabolites detected following the acute intake of purple wheat and purple barley products [5,7]. However, these previous studies reported peak urinary excretion within 0–4 h post-consumption [7], whereas in our study, maximum concentration was observed between 24 and 48 h. This pattern is consistent with the known colonic origin of hippuric acid, which would be expected to appear at later time points due to delayed absorption [21].

Flavan-3-ols and flavones also underwent A-ring cleavage, forming 1,3,5-trihydroxybenzene derivatives, while *O*-methylated flavones such as chrysoeriol generated methoxy-dihydroxybenzene compounds [22,23,24]. Due to analytical limitations, sulphated conjugates of 1,2,3- and 1,3,5-trihydroxybenzene could not be distinguished.

Finally, aromatic amino acids in WGB biscuits—tyrosine and phenylalanine (see Supplemental Appendix A)—may have contributed to this metabolite profile. Tyrosine can be converted to 3-(4-hydroxyphenyl)propanoic acid, while phenylalanine is a precursor of phenylacetic acid and hippuric acid [25], highlighting a complementary origin beyond polyphenols.

## 4. Conclusions

This study provides the first comprehensive characterization of the bioavailability and metabolism of (poly)phenols from processed purple whole grain barley (WGB) in humans. A total of 92 phenolic metabolites were identified in plasma and urine following acute intake, including native phase II conjugates and a broad array of microbial-derived catabolites.

Plasma analysis revealed early-phase absorption of anthocyanins, flavones, and hydroxycinnamic acid derivatives, while urinary data reflected extensive colonic metabolism of multiple phenolic families. The most abundant urinary metabolites included valerolactone conjugates, phenylpropanoic acids, and hippuric acid, supporting the key role of gut microbiota in the metabolic fate of WGB (poly)phenols.

Together, these findings highlight the dual origin—host and microbial—of circulating and excreted phenolic metabolites and expand current knowledge on the bioavailability of cereal phenolics. Given the scarcity of human data on pigmented cereals, the present study contributes new insights into the bioavailability and metabolic fate of barley-derived (poly)phenols, supporting their potential role in gut and metabolic health. Nevertheless, as this proof-of-concept study involved a single barley genotype, the results should be considered exploratory. Validation in larger cohorts and across diverse pigmented barley lines will be essential to confirm and extend these observations.

## Figures and Tables

**Figure 1 nutrients-17-03086-f001:**
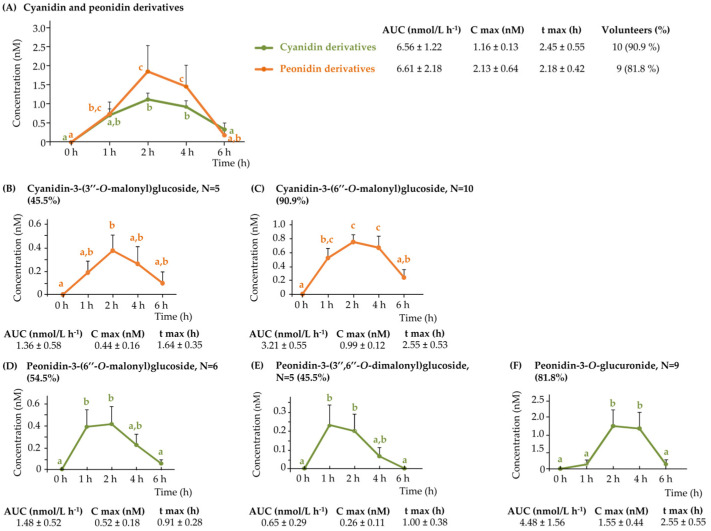
Pharmacokinetic profile for the detected anthocyanins in plasma samples following the acute intake of purple WGB biscuits: (**A**) cyanidin and peonidin derivatives, and the individual anthocyanins (**B**) cyanidin-3-(3″-*O*-malonyl)glucoside, (**C**) cyanidin-3-(6″-*O*-malonyl)glucoside, (**D**) peonidin-3-(6″-*O*-malonyl)glucoside, (**E**) peonidin-3-(3″,6″-O-dimalonyl)glucoside, and (**F**) peonidin-3-*O*-glucuronide.

**Figure 2 nutrients-17-03086-f002:**
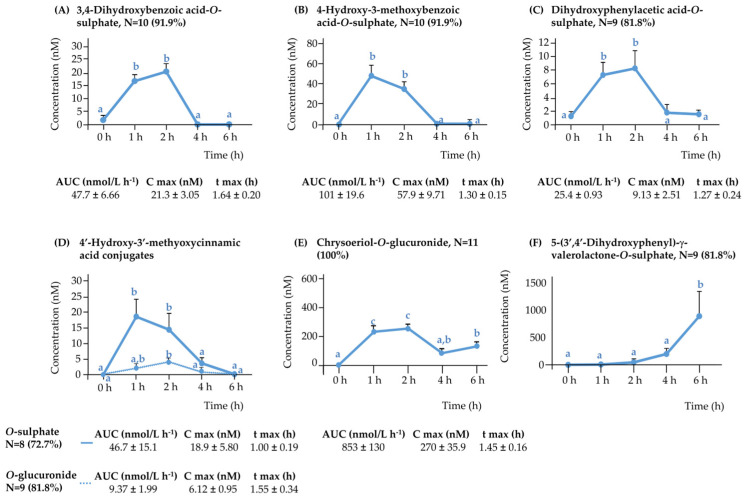
Pharmacokinetic profile for the detected rest of (poly)phenolics in plasma samples from 0 to 6 h following the acute intake of purple WGB biscuits: (**A**) 3,4-dihydroxybenzoic acid-*O*-sulphate, (**B**) 4-hydroxy-3-methoxybenzoic acid-*O*-sulphate, (**C**) dihydroxyphenylacetic acid-*O*-sulphate, (**D**) 4′-hydroxy-3′-methoxycinnamic acid conjugates, (**E**) chrysoeriol-*O*-glucuronide, and (**F**) 5-(3′,4′-dihydroxyphenyl-γ-valerolactone-*O*-glucuronide.

**Figure 3 nutrients-17-03086-f003:**
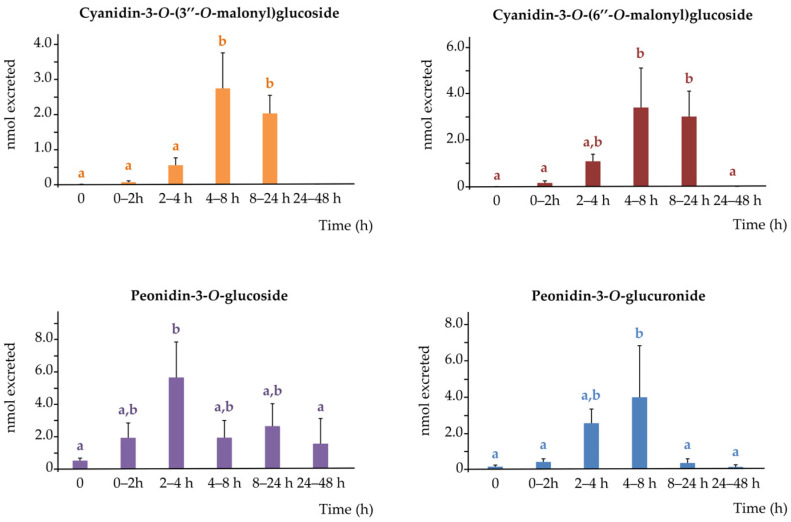
Urinary excretion of anthocyanins (nmols) at different time intervals from 0 to 48 h of the four anthocyanins detected.

**Figure 4 nutrients-17-03086-f004:**
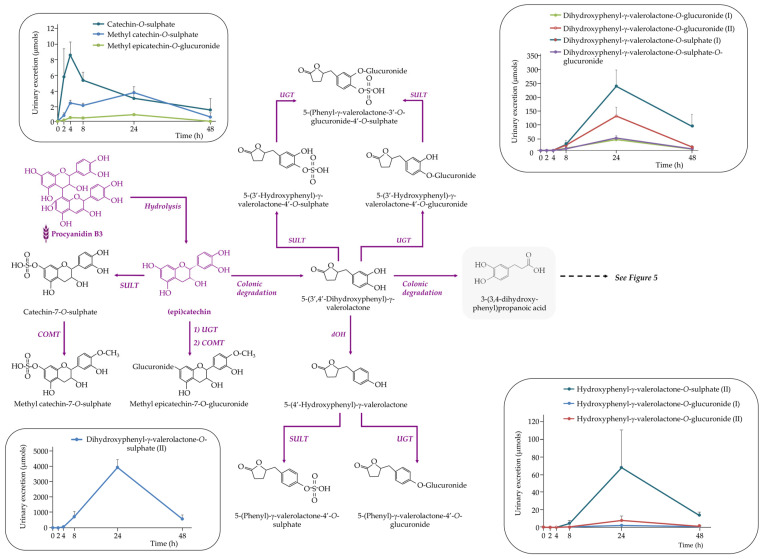
Proposed metabolic pathway for the generation of detected (poly)phenolic metabolites derived from flavan-3-ols following the acute intake of purple WGB biscuits.

**Figure 5 nutrients-17-03086-f005:**
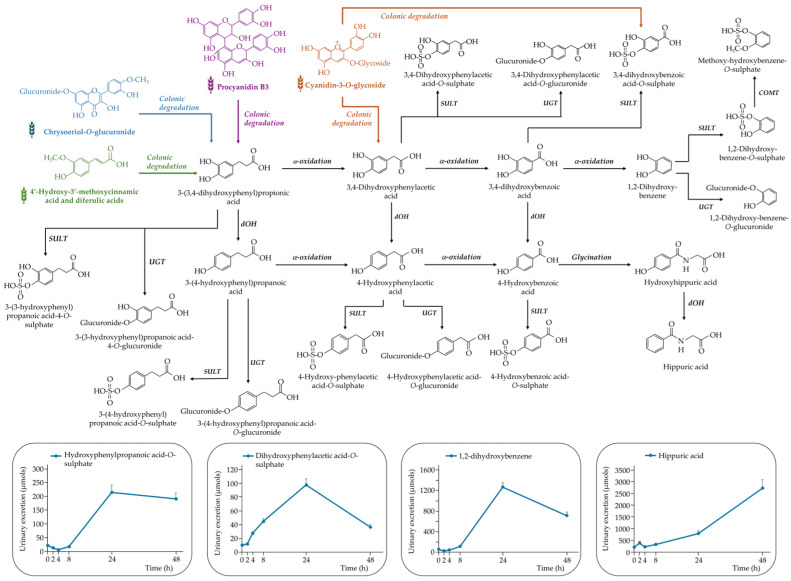
Proposed metabolic pathway for the generation of non-specific colonic (poly)phenolic-derived metabolites detected following the acute intake of purple WGB biscuits.

**Figure 6 nutrients-17-03086-f006:**
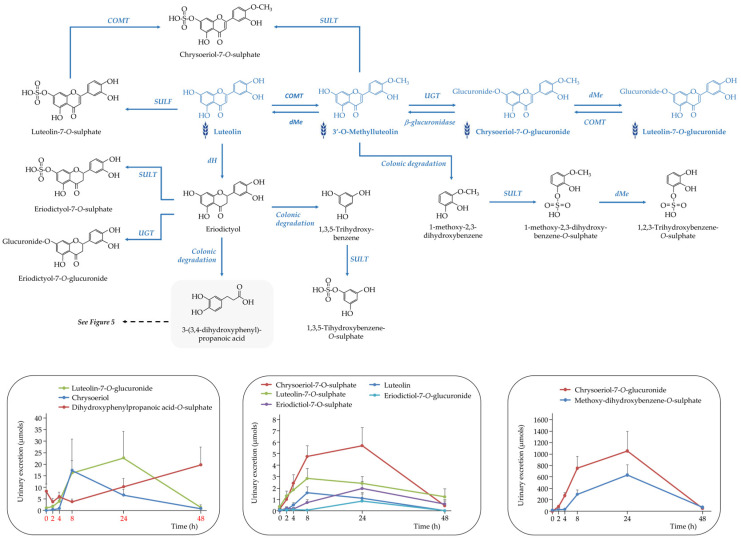
Proposed metabolic pathway for the generation of detected (poly)phenolic metabolites derived from flavones following the acute intake of purple WGB biscuits.

**Figure 7 nutrients-17-03086-f007:**
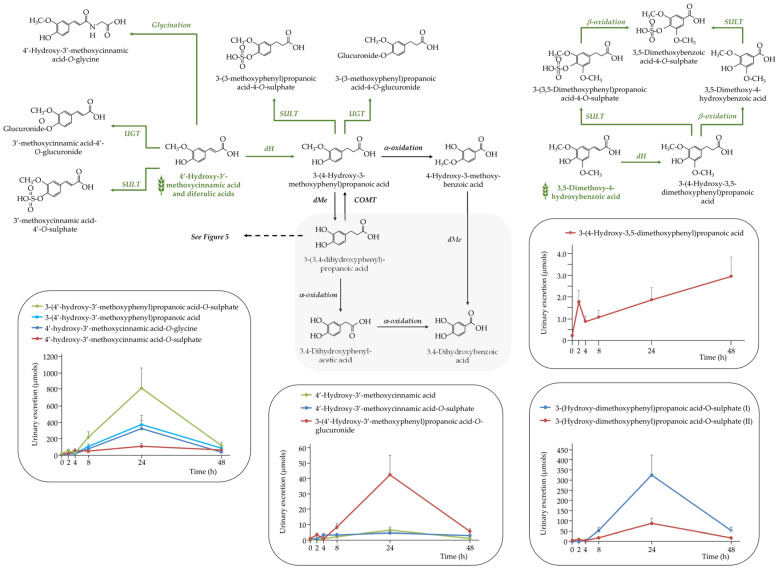
Proposed metabolic pathway for the generation of detected (poly)phenolic metabolites derived from 4′-hydroxy-3′-methoxycinnamic acid and its derivatives following the acute intake of purple WGB biscuits.

**Table 1 nutrients-17-03086-t001:** Free and bound (poly)phenolics (mg) in 140 g of WGB biscuits. Data is expressed as mean values ± standard deviation (SD). The number of replicates was three (*n* = 3).

	FREE (Poly)phenolics	BOUND (Poly)phenolics
Cyanidin-3-*O*-glucoside	6.20 ± 0.24	0.06 ± 0.01
Cyanidin-3-*O*-(3″-malonyl)glucoside	4.90 ± 0.20	n.d.
Cyanidin-3-*O*-(6″-malonyl)glucoside	12.5 ± 0.50	n.d.
Cyanidin-3-*O*-(3″,6″-dimalonyl)glucoside	16.9 ± 0.64	n.d.
Peonidin-3-*O*-glucoside	0.38 ± 0.07	0.02 ± 0.00
Peonidin-3-*O*-(3″-*O*-malonyl)glucoside	0.31 ± 0.01	n.d.
Peonidin-3-*O*-(6″-*O*-malonyl)glucoside	1.12 ± 0.07	n.d.
Peonidin-3-*O*-(3″,6″-dimalonyl)glucoside	0.46 ± 0.01	n.d.
Pelargonidin-3-*O*-glucoside	0.27 ± 0.02	0.02 ± 0.00
Pelargonidin-3-*O*-malonylglucoside	0.22 ± 0.02	n.d.
Pelargonidin-3-*O*-(6″-malonyl)glucoside	1.24 ± 0.08	n.d.
Delphinidin-3-*O*-glucoside	0.20 ± 0.06	n.d.
Total anthocyanins	44.7 ± 1.80	0.10 ± 0.01
4-Hydroxybenzoic acid	0.22 ± 0.02	0.87 ± 0.03
Hydroxybenzoic acid	n.d.	0.18 ± 0.01
3,4-Dihydroxybenzoic acid (PCA)	0.16 ± 0.06	0.03 ± 0.01
4-Hydroxy-3-methoxybenzoic acid (VA)	0.16 ± 0.03	0.44 ± 0.01
4-Hydroxy-3,5-dimethoxybenzoic acid (Syr)	n.d.	0.16 ± 0.02
Cinnamic acid	n.d.	0.11 ± 0.01
4′-Hydroxycinnamic acid	0.10 ± 0.02	1.18 ± 0.04
3,4-Dihydroxycinnamic acid (CA)	n.d.	0.02 ± 0.00
4′-Hydroxy-3′-methoxycinnamic acid (FA)	1.22 ± 0.14	40.8 ± 0.80
3′-Hydroxy-4′-methoxycinnamic acid (IsoFA)	0.20 ± 0.07	8.64 ± 0.17
4′-Hydroxy-3,5-dimethoxycinnamic acid	n.d.	3.07 ± 0.16
4′-Hydroxy-3,5-dimethoxycinnamic acid-*O*-glucoside	n.d.	0.95 ± 0.25
Diferulic acid	n.d.	12.4 ± 0.65
Diferulic acid DC	n.d.	1.20 ± 0.24
Triferulic acid	n.d.	2.06 ± 0.39
Total phenolic acids	2.06 ± 0.16	72.1 ± 1.24
Catechin	2.62 ± 0.56	n.d.
Catechin glucoside	2.09 ± 0.20	n.d.
Procyanidin B3	8.40 ± 1.49	n.d.
Gallocatechin-catechin or prodelphinidin B4	3.67 ± 0.39	n.d.
Total flavan-3-ols	16.8 ± 1.40	n.d.
Apigenin-7-*O*-glucoside	n.d.	0.04 ± 0.01
Apigenin-6-*C*-arabinoside-8-*C*-glucoside	0.46 ± 0.09	0.05 ± 0.01
Isovitexin-*C*-glucoside	0.56 ± 0.05	0.09 ± 0.01
Isovitexin-*C*-rutinoside	0.25 ± 0.03	n.d.
Isoscoparin-*C*-glucoside	0.89 ± 0.12	0.09 ± 0.02
Isoscoparin-*C*-rutinoside	0.26 ± 0.04	n.d.
Luteolin	5.92 ± 0.18	0.03 ± 0.00
Methyl luteolin (Chrysoeriol)	16.2 ± 2.63	0.07 ± 0.02
Luteolin-*O*-glucoside	0.13 ± 0.02	n.d.
Luteolin-7-*O*-glucuronide	32.9 ± 5.91	0.46 ± 0.03
Methyl Luteolin-*O*-glucoside	0.22 ± 0.03	n.d.
Methyl Luteolin-*O*-glucuronide	160 ± 24.6	4.64 ± 0.49
Total flavones	217 ± 33.0	5.53 ± 0.50
Total (poly)phenols	281 ± 32.4	77.7 ± 0.73
Total (poly)phenols (Free and Bound)	359 ± 32.2	

n.d., not detected.

## Data Availability

The original contributions presented in this study are included in the article and Appendix A. Further inquiries can be directed to the corresponding author.

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
