# Peer review of "Bioavailability and Metabolic Fate of (Poly)phenols from Hull-Less Purple Whole-Grain Barley in Humans"

_nutrients, 2025, doi:10.3390/nu17193086_

Round 1
Reviewer 1 Report
Comments and Suggestions for Authors
per attached file

Author Response
|
3. Point-by-point response to Comments and Suggestions for Authors |
|
1) SCIENCE: This manuscript is quite well organized and well written. However, several points require some clarification, improvement, or correction. Comments 1: 103: Description of the study participants: “A total of 14 healthy participants (8 women and 6 men”. This is contradicted by subsequent mentions of only 11 participants. This discrepancy needs some form of resolution or explanation. i. 327: “observed in 8 out of 11 participants” ii. Figure S1 describes the cohort as consisting of 6 women and 5 men for a total of 11. |
|
Response 1: We thank the reviewer for noticing this inconsistency. The correct number of participants was 11 (6 women and 5 men). The original mention of 14 participants at line 103 was an error, which has now been corrected in the revised manuscript (page 3, line 110). As noted by the reviewer, the information provided at line 327 (line 348 in the new manuscript) and in Figure S1 was already accurate and remains unchanged. |
|
Comments 2: 132: “μElution plates… were activated with 250 μL of methanol and equilibrated with 250 μL of 0.2% acetic acid.” It is not obvious to this reader what the intended meaning is. |
|
Response 2: We thank the reviewer for pointing out this ambiguity. To improve clarity, we have revised the sentence to read: “μElution plates were first conditioned with 250 μL of methanol and then equilibrated with 250 μL of 0.2% acetic acid to activate the sorbent prior to sample loading.” This revised wording has been incorporated in the manuscript (page 3, lines 140–142). Comments 3: 133: “centrifugation of 350 μL of venous plasma and 350 μL of 4% phosphoric acid”. Is the intended meaning that the plasma was added to 4% phosphoric acid prior to centrifugation? If so, the sentence could be revised to: “The 350 μL venous plasma was added to 350 μL phosphoric acid and then centrifuged at…” i. Other similar instances are on lines 143 and 144. Response 3: We appreciate the reviewer’s observation. To improve clarity, the sentence has been revised to: “A volume of 350 μL of venous plasma was mixed with 350 μL of 4% phosphoric acid and subsequently centrifuged at 8784 g for 10 min and 4 ºC”. This clarification has been incorporated into the manuscript (page 4, lines 142-144). Similarly, the description for the preparation of urine samples has also been clarified (page 4, lines 152–155). Comments 4: 317: While referring to Supplementary Figure S4B, the authors state that there was “similar total recovery for all four metabolites”. Yet looking at the presented data, it seems clear to this reader that the cumulative urinary excretion of Peonidin-3-O-glucoside is more than twice that of the other three metabolites. Are the authors suggesting that this difference is not significant? If so, it would be helpful to provide the p-value in the figure legend. Response 4: We thank the reviewer for this valuable observation. While the cumulative urinary excretion of Peonidin-3-O-glucoside appears higher than that of the other anthocyanin metabolites, statistical analysis (one-way ANOVA) revealed that the difference was not significant (p = 0.246). To clarify this, we have revised the legend of Supplementary Figure S4B to include the corresponding p-value (page 16, line 523).
Comments 5: 406: The authors state that “As shown in Figure 6, urinary chrysoeriol levels decreased notably at 8 h post-intake”. Yet, it seems clear from Figure 6 that urinary chrysoeriol does not substantially rise until at least the 20-hour mark, reaching a maximum at 30 hours. It is not clear to this reader what the authors are referring to. Response 5: We sincerely thank the reviewer for this valuable observation. The apparent inconsistency was due to a labeling error on the x-axis of Figure 6 (excretion time, h). Specifically, the time points were incorrectly reported: 10 h should correspond to 2 h, 20 h to 4 h, 30 h to 8 h, and 40 h to 24 h. We have now corrected the axis labels to accurately reflect the experimental time points, which resolves the discrepancy noted by the reviewer.
2) PRESENTATION: This manuscript is generally well written, with only a few minor needed corrections. Some examples are below. Comments 6: The figures are very information dense, and need to be made larger, with high resolution, to be more accessible to the reader. This is particularly the case for Figures 4, 5, 6 and 7. These figures include molecular structures of the various metabolites. The structures and labels, as they now stand, are not easy to see even after zooming in. Response 6: We sincerely thank the reviewer for this constructive comment. We agree that Figures 4–7 were overly information-dense, and that the resolution and size of the molecular structures and labels limited their readability. To address this, we have revised the figures by increasing both their size and resolution, and by enhancing the clarity of the molecular structures and labels. We believe these improvements substantially enhance the accessibility and readability of the figures for the reader.
i. Figures 4-7 have the molecular structures of barley polyphenols presented on top of images of growing barley. While the intention is lovely, the resulting overlay makes an already difficult to see structure even more difficult. It may be helpful to designate such structures with a smaller image, like a single barley sheaf, to one side of it. We thank the reviewer for this helpful suggestion. As recommended, we have modified Figures 4–7 by presenting the molecular structures on a clearer background and placing a smaller image of a barley sheaf to the side. This improves the visibility and readability of the structures.
Comments 7: 182: Change “post-consumption the WGB” to “post-consumption of the WGB” Response 7: Thank you for pointing this out. “Post-consumption the WGB” has been corrected to “post-consumption of the WGB” on page 5, line 200.
Comments 8: 197: Change “content in in 140 g of” to “content in 140 g of” Response 8: We thank the reviewer for noting this typographical error. The duplicated preposition has been removed, and the sentence now reads “content in 140 g of …” (page 6, line 208). Comments 9: 379: Change “sulphate and glucuronide conjugated.” to “sulphate and glucuronide conjugated variants.” Response 9: We thank the reviewer for this suggestion. The sentence has been revised to read “sulphate and glucuronide conjugated variants” in the manuscript (page 12, line 405). Comments 10: Supplemental page 5: Change “Supplemental Table s3” to “Supplemental Table S3 Response 10: We appreciate the reviewer’s observation. The letter “S” has been added to the caption of Supplemental Figure 3, now correctly labelled as “Supplemental Figure S3”.
|

Reviewer 2 Report
Comments and Suggestions for Authors
The paper is well written and easy to understand. However, the study contain mainly a very detailed description of polyphenols composition in one pigmented barley variety, biscuits from it and in blood samples and uninary samples after eating the pigmented barley from 11 people who has consumed the biscuits. The authors state that this is the first study in its kind. Basically the representation of both number of barley genotypes and study objects are low. And this is an issue, as the results (although containing a high number of analyses) cannot be seen as representative for people eating pigmented barley. This can just be seen as a just for this case picture. This has to be lifted and discussed in the paper, especially as the authors states that there is no earlier study to compare with. However, there is quite a number of earlier studies on other pigmented cereals, especially on wheat. And there are also studies on pigmented barley + wheat (e.g. Nignpense et al). Thus, the results from the present study need to be discussed in a wider context than in the present version of the paper. Wheat and barley are relatively closely related and share many features, I guess also polyphenols pathways. If not, that has to be discussed. This, introduction and discussion parts as well as conclusions and abstract need to be thoroughly revised.
Author Response
Response 1:
We sincerely thank the reviewer for this valuable comment. As noted, the available literature on the bioavailability of (poly)phenols following acute intake of pigmented cereals is very limited. To our knowledge, only two human studies have addressed this question: one by Gamel et al. (2019), who investigated purple wheat, and another by Nignpense et al. (2024), who examined purple wheat, blue wheat and purple barley.
Gamel et al. (2019) analyzed plasma and urine samples from 16 healthy participants after consumption of bran-enriched purple wheat products (bars and crackers). No intact anthocyanins were detected in plasma, and only three anthocyanin-related metabolites were identified in urine, with their structures remaining unidentified. For other phenolics, ferulic acid (plasma and urine) and hippuric acid (plasma) were the most abundant metabolites, while several additional compounds could not be structurally elucidated. These limited findings may reflect the relatively low phenolic content of the tested products, as the purple wheat bars and crackers contained approximately seven-fold lower concentrations of anthocyanins and about 1.5-fold lower levels of other phenolic compounds compared with the barley biscuits analyzed in our study. In addition, the analytical approach (HPLC-DAD with confirmation by a single MS detector) likely restricted metabolite identification.
The study by Nignpense et al. (2024) also presents important limitations. The trial included only three volunteers, with plasma and urine samples collected for a maximum of four hours post-consumption. Furthermore, the scope of metabolite identification was restricted, as only five metabolites were reported, two of which remained unidentified; the identified compounds were limited to protocatechuic acid, caffeic acid and hippuric acid.
In line with the reviewer’s suggestion, we have expanded both the discussions to include this previous evidence from wheat and barley. These revisions help to situate our findings within the broader context of pigmented cereals (page 2, lines 55–61; page 7, lines 262–267; page 11, lines 355–358; page 15, lines 480–485). The reference to Gamel et al. (2019) has been added to the References section.
Finally, we recognize the limitations of our study, particularly the focus on a single barley genotype. With respect to the sample size, we note that the inclusion of 11 healthy volunteers is consistent with standard practice in acute postprandial bioavailability trials, where the complexity and intensity of the experimental protocol usually limit recruitment. Nonetheless, we agree that our results should be interpreted as proof-of-concept, exploratory data that require validation in larger cohorts and across multiple pigmented barley lines. This revision underscores both the novelty of our work and the need for future research. As suggested, we have revised both the Abstract (Page 1, lines 26-28) and the Conclusions (Page 15, lines 503-506) to explicitly highlight these limitations and to frame our findings as proof-of-concept, exploratory data that require validation in larger cohorts and with additional pigmented barley genotypes.
References:
TH Gamel, AJ Wright, AJ Tucker, M Pickard, I Rabalski, M Podgorski, N Di Ilio, C O'Brien, EM Abdel-Aal. Absorption and metabolites of anthocyanins and phenolic acids after consumption of purple wheat crackers and bars by healthy adults. Journal of Cereal Science, 86 (2019) 60-68.
BE Nignpense, N Francis, C Blanchard, A Santhakumar. The bioavailability of polyphenols following acute consumption of pigmented barley and wheat. Food & Function, 15 (2024) 9330-9342.

Round 2
Reviewer 1 Report
Comments and Suggestions for Authors
Summary:
The authors revised the manuscript quite thoroughly. As a result, the manuscript has been substantially improved.
Feedback:
- Science:
The authors have carefully corrected and clarified the needed information.
- Presentation: Very minor correction needed.
- Figures 4 and 5: Now that the figures are clearer and legible, a typo has become visible. Change “Procyanidn B3” to “Procyanidin B3”.
Author Response
- Science:
The authors have carefully corrected and clarified the needed information.
Thank you very much for your valuable contribution
- Presentation: Very minor correction needed.
- Figures 4 and 5: Now that the figures are clearer and legible, a typo has become visible. Change “Procyanidn B3” to “Procyanidin B3”. According to the reviewer, the typopgraphic error has been corrected. In Figures 4 and 5, “Procyanidn B3” has been modified to “Procyanidin B3”